# Meldrum’s Acid Furfural Conjugate MAFC: A New Entry as Chromogenic Sensor for Specific Amine Identification

**DOI:** 10.3390/molecules28186627

**Published:** 2023-09-14

**Authors:** Lisa Zeußel, Sukhdeep Singh

**Affiliations:** 1Department of Nanobiosystem Technology, Institute of Chemistry and Biotechnology, Technical University Ilmenau, Prof-Schmidt-Straße 26, 98693 Ilmenau, Germany; lisa.zeussel@tu-ilmenau.de; 2Research Group Bioorganic Chemistry of Bioactive Surfaces, Institute of Chemistry and Biotechnology, Prof-Schmidt-Straße 26, 98693 Ilmenau, Germany

**Keywords:** amine sensing, Meldrum’s acid furfural conjugate, donor–acceptor Stenhouse adduct, optical sensor, molecular sensor

## Abstract

Bioactive amines are highly relevant for clinical and industrial application to ensure the metabolic status of a biological process. Apart from this, generally, amine identification is a key step in various bioorganic processes ranging from protein chemistry to biomaterial fabrication. However, many amines have a negative impact on the environment and the excess intake of amines can have tremendous adverse health effects. Thus, easy, fast, sensitive, and reliable sensing methods for amine identification are strongly searched for. In the past few years, Meldrum’s acid furfural conjugate (MAFC) has been extensively explored as a starting material for the synthesis of photoswitchable donor–acceptor Stenhouse adducts (DASA). DASA formation hereby results from the rapid reaction of MAFC with primary and secondary amines, which has so far been demonstrated through numerous publications for different applications. The linear form of the MAFC-based DASA exhibits intense pink coloration due to its linear conjugated triene-2-ol conformation, which has inspired researchers to use this easy synthesizable molecule as an optical sensor for primary, secondary, and biogenic amines. Due to its new entry into amine identification, a collection of the literature exclusively on MAFC is demanded. In this mini review, we intend to present the state-of-the-art of MAFC as an optical molecular sensor in hopes to motivate researchers to find even more applications of MAFC-based sensors and methods that pave the way to their usage in medicinal applications.

## 1. Introduction

After carbon and oxygen, nitrogen is a dominating heteroatom that has a significant role in molecular science. The presence of nitrogen can be witnessed in various biomolecules like DNA, RNA, protein, and even polysaccharides. In simple terms, the nature of chemical bonds could be amide or amine depending upon the neighboring atoms or group of atoms. When it comes to the reactivity, amine are for sure more reactive than amides, and therefore, are abundantly used functional groups that finds its application in medicine, fertilizer, and pesticide production, protein quantification, in the petroleum and electrical industries, as well as in environmental protection [1]. In the food industry, amine sensing is further employed for the detection of food spoilage, which is characterized by the release of volatile or biogenic amines, e.g., ammonia, dimethylamine, trimethylamine, histamine, putrescine, cadaverine, spermidine, and spermine [2]. Such volatile amines can cause severe health issues due to their extreme toxicity when consumed by humans [2,3], and thus, the identification of amines in a sample is highly relevant. Various excellent review articles have summarized the use of molecular probes and sensors for the efficient detection of biogenic and organic amines [2,4,5,6]. 

One of the oldest detection agents for amines is ninhydrin, which was already discovered in 1910 by Siegfried Ruhemann. He found that ninhydrin reacts with primary amines to form a deep blue compound named Ruhemann’s purple [7]. Today ninhydrin-based reagents are used in numerous laboratories for forensic fingerprint detection [8,9], food chemistry [10], clinical chemistry [11], microbiology [12], pharmacology [13], or toxicology [14]. However, its use poses some disadvantages due to pH dependency, non-stoichiometric color formation, or low selectivity in more complex analytic solutions [15]. Moreover, the chromophore is not chemically bound to the substrate and remains stable even when the substrate is removed, which could lead to false positive results [15]. Therefore, excessive sample manipulation and/or purification, e.g., ion exchange chromatography is needed prior to ninhydrin application, when specific amine-containing substances are to be detected. Other approaches of ninhydrin amine sensing include expensive instrumental setups like liquid chromatography/tandem mass spectrometry (LS/MS/MS), which further requires trained staff for the operation and analysis [16]. Various techniques have been developed based on different detection agents and sensing methods. But often, these techniques involve not only expensive equipment but also difficult on-site analysis due to poor portability, tedious analysis procedures, or low selectivity and specificity [17,18]. Thus, new inexpensive methods providing facile operation, rapid analyte quantification, and high sensing capacity are always searched for. 

Today, most abundant sensors for amine detection are based on optical sensing methods; which offer easy operation, rapid reaction times, and good sensitivity and selectivity. However, most often, analytes miss the typical structural features, e.g., conjugated π-systems, which are needed for optical detection. Therefore, optical sensing systems in general include molecules that react with the analyte solution to optically active derivatives that provide the possibility of optical measurements [3]. The analysis can be performed by measuring various parameters depending on the chosen type of detecting agent, e.g., absorbance, fluorescence, (chemi-)luminescence, energy transfer, reflectance, light scattering, refractive index, diffraction, or polarization [2,3]. Especially for the development of device-based amine sensors, an enhancement of fluorescence intensity, the so-called “turn-on”, is desirable. Kumpf et al. has demonstrated the use of extended distyrylbenzenes as a strip-based “turn on” assay [19]. Mani et al. developed a zinc-based coordination polymer for the selective “turn on” detection of aliphatic amines [20]. Synthetically simple di-catechol (dicat) have shown to reach the detection limit down to the sub ppm level [21]. Similarly, 1,4-diazine-based dyes have shown a selective chemosensing ability towards aliphatic amines [22]. Enzymes, antibodies, molecularly imprinted polymers (MIPs), and aptamer-based biogenic amine sensors are dominating the literature [23]; however, single molecule-based sensors are much more simple to synthesize. In our opinion, the development of a chemosensor with the best performance has to consider a delicate balance between selectivity, the limit of detection, and simple analytics, and it should also be economical. There, colorimetric sensing is rather advantageous because of the naked eye detection feature, where basically no expensive instrumentation is required. 

Chemosensing using small organic molecules are traditional as well as efficient. Therefore, various new motifs are screened for their sensing applications. Among many, activated furans appear to be good possible candidates for such a venture. Generally, furan offers various applications in synthesis, drug discovery, and photolithography, and its derivatives have been extensively explored in sensor application due to its elite chemistry [24]. In this mini-review, we are showcasing a newly emerging activated furan, Meldrum’s acid furfural conjugate (MAFC), or otherwise called MAF, as a colorimetric detecting agent for primary and secondary amines, which can be easily synthesized via the Knoevenagel condensation of simple and inexpensive precursors, 2-furfural and Meldrum’s acid [25]. MAFC offers advantages over other systems due to its easy synthesis, fast reaction times, solubility in benign solvents, and high absorption coefficient (ε_λmax_ ≈ 10^5^ M^−1^ cm^−1^) [26]. Similarly, activated furans, e.g., the barbituric acid activated furan conjugate (BAFC), are also capable of generating intense coloration upon its reaction with amines. However, its scarce solubility in benign solvents like ethanol limits it applicability as a colorimetric sensor. Therefore, despite both being activated furan in nature, MAFC has overtaken BAFC in sensing applications. The utility of MAFC as a biogenic amine sensor is increasing day by day due to its easy synthesis and facile operation. An intricate ring opening color chemistry enhances the role of MAFC in the amine-sensing literature. Therefore, in this mini review, we complied literature on the use of MAFC as a molecular sensor for amine detection to highlight its potential in the field of chemosensory. 

## 2. Furan Derivatives as Detection Agents

Before describing the potential of activated furan as selective amine sensors, it is desirable to showcase the general potential of furan-based chemical systems in chemosensory application. Electron rich furan oxygen atoms contributes to both the aromatic ring character as well as dative bond formation. This particular molecular characteristic is responsible for the utility of furan-based probes, especially for trace element sensing applications. Efficient and selective methods for the colorimetric detection of trace elements and substances are the bases of modern assay development [27]. Especially, fast, easy, and selective sensing of various ions with a low limit of detection (LOD) is of the highest relevance, due to their importance in biology, medicine, and environmental protection [28]. Thus, research groups synthesized various furan derivatives those provided methods of the fluorometric sensing of metal ions via mechanisms like the intramolecular charge transfer (ICT) (see Figure 1). Cu^2+^ is the third most abundant essential transition metal ion in the human body and participates in multiple tasks, e.g., as a cofactor for electron transport or a catalyst in oxidation-reduction reactions. However, an excess of Cu^2+^ intake can be harmful to humans and its maximum limit in drinking water is therefore limited to 20 µM [29]. Cu^2+^ has a natural fluorescence quenching behavior, which makes the fluorometric “Turn-off” of Cu^2+^ sensing a convenient and easy-to-apply method [29]. However, other published methods also demonstrate the possibility of the detecting presence of Cu^2+^ ions with other mechanisms. Thereby, a “Turn-on” approach with an emerging fluorescence emission at 394 nm [30] as well as the measurement of the bathochromic shift of the fluorescence emission from 550 nm to 600 nm in response to the complexation of Cu^2+^ to a furan-derivatized BODPIY sensor were demonstrated [31]. F^-^ intake is another factor that needs detailed monitoring due to the ions’ relevance in dental health. The limit of F^−^ ion concentration in drinking water is set to 79 µM by the World Health Organization (WHO), which obviously requires regular, easy, fast, and reliable on-site testing [32]. Two furan-containing fluorometric sensors for this application were demonstrated based on julolidine [32] and naphtalimide [33]. Ratiometric sensing of Zn^2+^ was shown in 2017 with 7-diethylamino-3-formyl-coumarin-(2′-furan-formyl)-hydrazone via Zn^2+^ complexation, which induced a decrease in fluorescence emission at 511 nm and an increase at 520 nm when excited with 322 nm of light [34]. Al^3+^ was realized in a different study with various bishydrazide compounds, which exhibit strong fluorescence emission at 467 nm, 524 nm, and 601 nm, respectively, due to their metal chelation-enhanced fluorescence [35]. Lastly, the sensing of Cr^3+^, an essential trace element in food and nutrient supplements, was demonstrated with 5-(furan-2-yl)-7,8,13,14-tetrahydrodibenzo[a,i]phenanthridine synthesized via the condensation reaction of 2-tetralone and furfural in the presence of ammonium acetate [36]. ICT results in a rapid strong increase in fluorescence emission at 405 nm were directly proportional to the Cr^3+^ concentration. When prepared as a paper test strip, this sensor can be used for the detection of Cr^3+^ in environmental water samples. However, besides (metal) ion detection, only few applications of furan derivatives in optical detection agents are known (except colorimetric methods). Examples can be found in the publications of Liabsungneons et al. [37] and Wakshe et al. [38], which demonstrate the sensing of 2,4,6-Trinitrotoluene and 4-nitrophenol with hydrazonefuran- or quinazolinone-based 2-(furan-2-yl)-2,3-dihydroquinazolin-4(*1H*)-one fluorescent organic nanoparticles, respectively. 

## 3. Activated Furans for Colorimetric Sensing

As depicted in Figure 1, furan-containing sensors mainly work via the complexation of analytes through either the furan/furfural moiety or other functional sites in the detection agent. In the latter case, furan substitutions were shown to enhance the selectivity of the sensor molecule [33]. The sensing of specific amines was not demonstrated via these approaches. However, in the particular case of activated furans, which means the conjugation of the furan ring to an electron-deficient molecular species, the furan ring-opening reaction in response to the interaction with donor amines is known to result in a high colored reminiscent of the triene-2-ol fragment. Being surrounded by an electron rich amine and electron deficient activator, the triene-2-ol fragment could photoisomerize to cyclopentenone, which is extensively explored as a photoswitching molecule (see Figure 2A). However, from a chemosensing point of view, the intensely colored triene-2-ol formation upon reaction with amine is a unique feature of this molecule for being explored as a colorimetric sensor of amines. 

The ring opening reaction of furan itself was first reported by John Stenhouse in 1850 via the reaction of aniline with a furfural-containing crude oil to the colorful Stenhouse salt (see Figure 2B) [39]. Acidic treatment of the Stenhouse salt leads to a conformation change of triene-2-ol to its colorless 4,5-diamino-2-cyclopentenone via a conrotatory 4π-electrocyclization, which then rearranges to its thermally more stable 2,4-isomer. The reaction did not find high interest at that time due to its low selectivity, and the chemistry community tried to find different ways for a more selective generation of Stenhouse salts via either varying reaction conditions or reactants. In 1976, the famous Piancatelli rearrangement was first demonstrated via the acid-catalyzed reaction of 2-furylcarbinols to 4-hydroxy-cyclopentenone. However, for this reaction to be possible, 2-furylcarbinol first needed to be generated from furfural via the Grignard reaction and the reaction conditions were highly dependent on educt reactivity. From this point forward, numerous groups investigated methods based on the Piancatelli rearrangement which could provide a way to synthesize the functional five-membered rings with less harsh conditions compared to early studies [24,40,41]. Such protocols were, e.g., used for the synthesis of natural products like heptemerone G from 2-furylcarbinols [24,42]. In 2007, Batey and Li also showed the successful synthesis of the natural product Agelastatin A directly from furfural with an excellent yield via dysprosium(III)-trifluoromethanesulfonate catalysis [43]. In 2014, the portfolio of activated furans was again expanded via the synthesis of the novel so-called donor–acceptor Stenhouse adducts (DASA) by the Read de Alaniz group [25]. Hereby, activated furans were generated via the Knoevenagel condensation of furfural and Meldrum’s acid or 1,3-dimethylbarbituric acid, respectively, which were classified as first-generation DASA. Colorful Stenhouse salt like DASA were then formed via the reaction of activated furans with primary or secondary amines. Both the synthesis of activated furan as well as the formation of DASA can be performed efficiently at mild conditions without the need for catalysis or harsh chemicals. Conformation change via linear to cyclic isomerization from colored triene-2-ol to colorless cyclopentenone conformation can, in general, easily be performed via visible light exposure and is usually reversible via moderate heat influence in the solution (see Figure 2A) [24]. Until today, three different generations of DASA have been defined which differ in their electron-donating and -accepting moieties. Every generation has its own advantages and disadvantages regarding solubility, reaction kinetics, and reversible photoswitching behavior (see Figure 2C) [44]. Since their first demonstration, DASA have been extensively used in various applications, ranging from drug delivery, optical sensing, phase transfer catalyst recovery, liquid crystal polymer materials, and wavelength-selective photoelectric switches to material surface modification [45]. Additionally, both triene-2-ol and cyclopentenone derivatives were explored for sensing purposes, e.g., for the detection of nerve agents [46], neurotransmitters [47], and metal ions [18,48,49], as shown in Figure 2D. In general, all activated furan which are conjugated to an electron deficient motif are capable of giving a rapid reaction with primary and secondary amines, which makes them ideal candidates for amine-sensing applications. However, among them, MAFC succeeds well in amine identification, due to its facile synthesis, robust shelf life, and ease of solubility.

**Figure 2 molecules-28-06627-f002:**
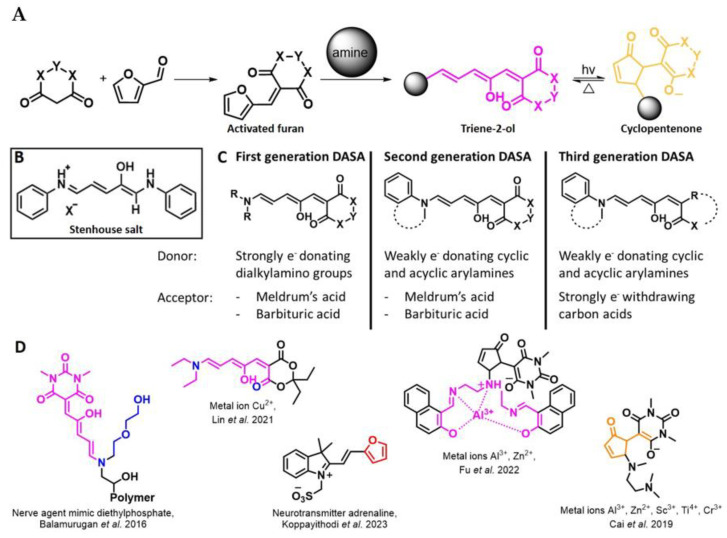
(**A**) Schematic drawing of activated furan synthesis, generation of colored linear DASA via reaction of activated furan with secondary amine and light-induced reversible linear to cyclic isomerization to colorless cyclopentenone conformation. (**B**) Molecular structure of Stenhouse salt. (**C**) Schematic comparison of molecular structure of the three DASA generations developed so far. (**D**) Molecular sensors based on activated furan and DASA derivatives [18,46,47,48,49]. Colored motifs depict the molecular scaffold of the sensor, which is actively participating in the sensing process.

## 4. Meldrum’s Acid-Activated Furan Conjugate as Amine Detecting Agent

MAFC is a first-generation DASA precursor and offers advantages in regard to the fast reaction kinetics with amines, high solubility in benign solvents, and good reversibility of visible light-induced linear to cyclic isomerization via moderate heat exposure in the solution as well as the vapor phase and specific solid-state systems. Further synthesis of activated furan MAFC can be readily performed by mixing 2,2-dimethyl-1,3-dioxane-4,6-dione and 2-furfural in water. Hereby, synthesis is so easy that Helmy et al. recently proposed to use this reaction for demonstrating photochromism to undergraduate practical courses [50]. Due to this and its extremely fast DASA formation with primary and secondary amines, MAFC recently gained more and more attention as a colorimetric sensor for amines in the solution, vapor, and solid phase [26].

It was demonstrated that DASA grafting and its photoswitching on a solid surface via the reaction of the secondary amine-rich polycarbonate surface with MAFC is possible. Obvious intense coloration, even on sterically restricted solid surfaces, promotes the use of MAFC as an amine indicator [51]. The initial report was soon followed by the first and currently highly cited reports on amine sensing in the solid, solution, and vapor phase with activated furan [26]. Diaz et al. [26] showed that MAFC reacts two times faster with diethylamine (DEA) than BAFC in tetrahydrofuran (THF). Solution phase reactions hereby were strongly dependent on the amine type with secondary amines resulting in the fastest reaction followed by primary amines and ammonia. The solid-state sensing capacity was examined via the application of MAFC in thin layer chromatography by successfully monitoring a tryptamine-based synthesis sequence. Further, the staining of amine-functional resins from the solid-phase synthesis of peptides and peptidomimetics could be shown. The sensing of amines in the vapor phase was first examined via the preparation of MAFC nylon filter membranes, which were sealed in septa-capped scintillation vials. Exposure to dimethylamine (DMA) and ammonia vapors resulted in the appearance of a distinct color for concentrations ≥0.5 ppm (see Figure 3A). As typical examples for volatile amines released from aging meat, this provided insight into the suitability of MAFC-based detection for food spoilage. Therefore, Diaz et al. [26] applied the MAFC nylon filter membranes in an experiment with fresh fish in which a clear appearance of color on the membranes over a 48 h experimental time indicated meat spoilage. The results provided evidence of the suitability of MAFC as a colorimetric sensor for food monitoring. Moreover, the method was investigated further regarding its reactivity towards primary, secondary, and tertiary amines [52]. Hereby, various linkers and terminal functional groups were created on polystyrene (PS) beads and reacted with a 0.1% ethanolic solution of MAFC. All amine functionalization except tertiary amines or bromo-, phenyl, or aliphatic chain-containing amines resulted in the appearance of a purple color, indicating successful DASA formation. Changes were observed between the primary and secondary amines such that primary amines led to the release of color from the PS surface into the solution while secondary amines resulted in the stable coloration of the PS surface. Additionally, the formation of primary amine DASA was dependent on the water content, and thus, no reaction could be observed when the 1:1 EtOH/water solution was used in contrast to pure EtOH. Since the secondary amine DASA formation was not influenced via the water addition, it could be demonstrated that MAFC can be used as a selective detection agent for secondary DASA on the PS surface (see Figure 3B). Moreover, a protocol was established for grafting primary, secondary, and tertiary amines on the polycarbonate (PC) surface to investigate DASA formation on the thermoplastic material, which is relevant for research in bioactive surfaces. Similar to PS, DASA formation on aminated PC was selective for secondary amines. Thus, a sensitive, rapid, reliable, and operationally simple method was demonstrated for secondary amine detection and DASA grafting on the PS and PC surface. As implied before, one constrain of MAFC-based amine sensing is its operation in an aqueous solution. The high polarity of water often leads to the unwanted linear to cyclic isomerization of first-generation DASA and can even induce the leaching of color into the solution in solid-state systems. Chen et al. found a way to circumvent these problems by reacting carboxylic acid-functionalized MAFC with oxa-norbornene to form an activated furan oxa-norbornene derivative (ON-MAFC) [53]. A MAFC-based polymer was then generated via ring-opening metathesis polymerization (ROMP) with the Grubbs initiator, ON-MAFC, and other oxa-norbornene monomers of various alkyl chain lengths from C2–C16. Spincoating and subsequent dip coating of the MAFC-based polymer in amine solutions for 3 min at 75 °C was employed for the investigation of the sensing capacity. The LODs found were 20 ppm, 10 ppm, 20 ppm, and 100 ppm for the aqueous diethylamine, n-butylamine, indoline, and p-methoxyaniline solutions, respectively. No color leaching or linear to cyclic isomerization was observed even when the samples remained in the aqueous amine solutions for a prolonged time (see Figure 3C). Additionally, the group found that the glass transition temperature (T_g_) is directly correlated with the response of the MAFC-based polymer. Even though such activated furan derivatized polymers do not allow for the selective sensing of primary or secondary amines in a mixture of those, this work opened the way for the use of activated furan-based amine sensing in aqueous conditions that can be controlled via T_g_ and the reaction temperature. Sensors based on activated furans are often solely used in the solution phase or simply coated on a solid substrate. A different approach for the preparation of solid-state DASA sensors was demonstrated via the electrospinning of polycaprolacton and polylactic acid mixed with activated furans [54]. Three different activated furans were investigated: MAFC, n-substituted BAFC, and a pyrazolone-based activated furan. Even though the pyrazolone-based activated furan, which is a representative for third-generation DASA, reacted rapidly with the minimal concentrations of DEA (LOD 10 ppb), it did not permit the quantification of low amine concentrations due to an early ΔRGB plateau. MAFC, on the other hand, could easily be applied for the quantification of volatile amines on the off-side of higher LOD (1 ppm). BAFC-containing electrospun meshes exhibited the worst performance and provided the LOD of 1–10 ppm (see Figure 3D). The electrospun meshes of activated furans or the combination of multiple activated furans in such meshes could thus be used for the quick testing of volatile amines in medical screening, workspace safety, or drug testing. If the quantification of amines is required, MAFC is by far the best candidate. The authors stated that with further improvement of the electrospun mesh detecting system applications as a colorimetric sensor for diagnosis in the exhaled breath, e.g., for the detection of pneumonia or chronic kidney disease, are potentially possible.

First real-world application possibilities of amine detection with MAFC were just recently demonstrated, where Sabahi-Agabager et al. used MAFC for the quantification of mesalazine (MES) in complex solutions containing surfactants, different ions, or organic chemicals and other pharmaceutical amines [55]. MES is a safe drug with very few adverse effects and good tolerances for the cure of Crohn’s and inflammatory bowel disease. Rapid reaction of MAFC with the MES primary amine functional group results in a strongly purple-colored product with a maximum absorption at 575 nm that fades over time due to the cyclization of the Stenhouse adduct. Extensive optimization was performed with a high variety of methods until the optimum conditions were found. These studies were performed on one side via practical experiments like UV-Vis or NMR spectroscopy, and on the other side, they were calculated via the density functional theory (DFT), time dependent DFT (TD-DFT), the univariate method, and the central composite design optimization method (CCD). 3.0% (wt/v) Triton X as the surfactant, 2 mM of MAFC and 30 mM of NaOAc for matrix normalization and pH stabilization, 0.5 mL of EtOH as the solvent, and a 10 min reaction time were found to be the optimal parameters for the proposed MES sensor. With those conditions, a LOD of 0.04 µg/mL for MES was achieved. DFT calculations revealed three possible structures of which the neutral form and the anionic form formed via dissociation were the most probable ones. In another instance, a real-world application was presented by Cho et al. in the field of forensic drug analysis [56]. On-site field tests for amphetamine-type stimulants like methamphetamine (METH) and 3,4-methylenedioxymethamphetamine (MDMA) in aqueous solutions, spiked drinks, or even tablets are highly necessary to be able to quickly grasp the danger of a situation. Methods like HPLC obviously are unsuitable for such tests due to their long analysis times, difficult operation, and fragile instruments. Known on-site field test agents, e.g., Marquis and Simon’s reagent, on the other hand, are limited in their use due to ambiguous responses in the presence of excipients, adulterants, or other interferences, as well as their toxic or corrosive nature. The authors demonstrated that MAFC quickly reacts with the secondary amines of METH and MDMA, resulting in strongly colored DASA in the solution (spiked drinks), as well as in the solid phase (ecstasy tablets) (see Figure 3E). The LOD for METH and MDMA was found to be 0.36 µg/mL and 0.57 µg/mL, respectively. The strong color is easy detectable with the naked eye, providing quick information of amphetamine presence without the need for excessive instrumental operation.

**Figure 3 molecules-28-06627-f003:**
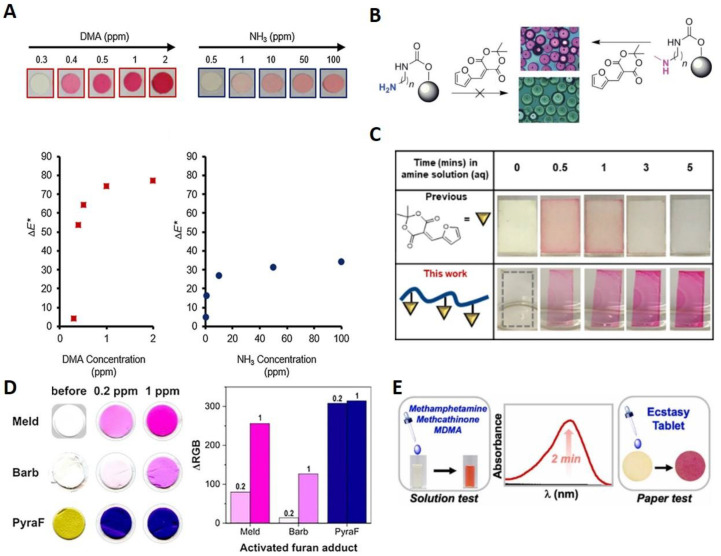
General studies with focus on amine sensing with MAFC. (**A**) Visual representation and corresponding data points of nylon filter membranes coated with MAFC after 5 min exposure to different concentrations of DMA and NH_3_. Copyright Wiley-VCH GmbH. Reproduced with permission from [26]. (**B**) Reaction of MAFC with primary and secondary amine functionalized polymer beads showing selective coloration of secondary amine-containing PS beads. Reprinted from [52]. (**C**) Demonstration of improvement of stability of covalently bound compared to non-covalently bound MAFC in aqueous conditions over period of 5 min. Reprinted with permission from [53]. (**D**) Visual representation and ΔRGB plateau of MAFC-, BAFC-, and pyrazolone-based activated furan electrospun meshes before and after reaction with 0.2 ppm and 1 ppm DEA, respectively. Reprinted from [54]. (**E**) Graphical depiction of MDMA and METH sensing possibility with MAFC in solution as well as solid phase. Reprinted with permission from [56].

Another field of application, where MAFC-based amine sensing gained recent importance, is amino acid and protein sensing for fields like biology or food monitoring. In our group, we investigated the sensing capabilities of MAFC towards specific amino acids. First, we studied the general sensing possibilities of MAFC towards 14 representative amino acids (A, R, C, E, G, H, L, K, M, F, P, W, Y, and V) at neutral pH [57]. Interestingly, we found that only the reaction of MAFC with amino acids lysine (K) and arginine (R) resulted in a prominent color formation attributable to DASA generation (see Figure 4A). Other amino acids, including amine-rich histidine (H), did not lead to obvious DASA formation. Interaction of MAFC with lysine was investigated in more detail, revealing that a successful reaction is achievable in solutions with pH 4–11 in a time frame of approximately 30 min. The LOD was found to be 100 µM. ^1^H-NMR and LCMS studies were performed to explore the lysine–DASA molecular structure, which was proposed to be a 1:1 adduct of lysine and MAFC in the trihydrate state. Detection possibility of lysine in complex samples was furthermore demonstrated by adding MAFC to a solution of pea protein. Distinct color appearance provided evidence of lysine in the sample. In a follow-up study, the pH dependence of amino acid sensing with MAFC was examined in more detail [58]. Whilst pH dependence in terms of lysine detection with MAFC is low (as long as 3 < pH < 12), it was demonstrated with UV-Vis analysis that other amino acids with amines included in the five-membered rings react stronger with MAFC at pH 11 compared to neutral conditions. However, with the naked eye, the successful reaction was only visible for amino acid proline (P). P resulted in exceptionally fast and strong color formation. Only Ps’ natural derivative 4-hydroxyproline showed an even more intense reaction with MAFC (see Figure 4B). Impressive LODs of only 11 µM and 6 µM, respectively, could be demonstrated, which is 10 times lower than for the previously reported lysine-sensing system. MAFC can therefore not only be used as a detecting agent for lysine and arginine at neutral pH but also as a selective detection agent for P in biological samples at pH 11. Recently, Ajayan et al. demonstrated the possibility of activated furan to quantify protein concentration via the colorimetric sensing with MAFC (see Figure 4C) [17]. MAFC was hereby dissolved in DMSO and either used directly for the sensing of hydrophobic proteins or in combination with proteins dissolved in aqueous solutions. The sensing capability was shown via the quantification of Bovine Serum Albumin. A LOD of 125 µg/mL within a 15–60 min sensing time was reported, whereby the absorbance increased proportional to the protein concentration. The fast reaction time combined with an inertness towards the detergents, ease of operation, and simple analysis is advantageous compared to other established methods like the Bradford assay or mass spectrometry. The high affinity of MAFC towards lysine also inspired another research group to involve MAFC in a high throughput screening system tailored to the most relevant topic of polymer recycling [59]. Enzymatic degradation was used to hydrolyze nitrogen-containing synthetic polymers, e.g., polyamides (PA) and polyurethanes (PUR). Lysine structural analogs 6-aminohexanoic acid and hexamethylenediamine as well as PUR degradation products 2,4-toluenediamine, 2,6-toluenediamine, and 4,4′-methylenedianiline were easily detectable after the addition of MAFC via the absorption peak at 494 nm with the LODs of 49 µM, 23.8 µM, 0.6 µM, 10.6 µM, and 5.9 µM, respectively. Sensitivity and accuracy of the colorimetric detection of degradation products were hereby comparable with the HPLC analysis results. Thus, colorimetric detection was successfully applied to investigate the suitability of applied enzymes and experimental parameters for PA and PUR degradation (see Figure 4D). This research therefore demonstrates the possibility of using MAFC amine sensing as an easy and cheap process control method.

The above-mentioned studies all involve the sensing of primary and secondary amines via the rapid reaction of MAFC with this chemical species, which was also the focus of this review. However, few other sensing possibilities exist that should not be neglected here. The return of coloration of DASA in response to heat exposure was soon found after their initial demonstration to be suitable for temperature-mapping sensors. This is relevant for the determination of armor, solid rocket motors, or explosive formulation. The formation of micro cracks in response to micron-scale temperature increases via impact or shock loading can tremendously influence the safety of such substances, and thus, such an influence needs to be monitored to ensure safe application. Dioctyl–DASA was generated via the reaction of MAFC with a dioctyl secondary amine donor and distributed in hydroxyl-terminated polybutadiene (HTPB), a common binder in explosive and rocket motor formulations [60]. Linear to cyclic isomerization was induced with 80 h of light exposure, resulting in a colorless dioctyl–DASA HTPB. The heating of such samples in a CDS analytical pyroprobe at 20 °C/ms to 120 °C for 30 min leads to a clear return of color attributable to triene-2-ol formation. UV-Vis absorption and fluorescence measurements reveal new peaks at 544 nm and 570 nm, respectively. A formed linear dioctyl–DASA HTPB was demonstrated to be stable in the dark over several days. Thus, the proposed thermal mapping sensor provides a thermal impact memory and no need for an ultrafast measurement of absorption and fluorescence. Practical applicability was demonstrated via bullet impact experiments by shooting a 7.62 mm NATO bullet via the colorless DASA functionalized polymer. The bullet path was clearly visible through the strong, localized pink color in the sample and the local temperatures were shown to be calculable with a specific formula. Further, MAFC-based DASAs were also investigated towards their application in metal ion sensing for Cu^2+^ and Fe^3+^. The detection is based on the metal ions’ inherent fluorescence-quenching characteristic. DASA polyethylene imine polymer dots (DASA–PEI) were synthesized via the reaction of MAFC with PEI, resulting in a strong absorption at 366 nm and 522 nm, as well as a fluorescence emission at 520 nm after an excitation at 420 nm [61]. Thereby, the absorption properties of DASA–PEI can be influenced with visible light exposure and pH change to alkaline pH 8 by inducing linear to cyclic isomerization to a yellow cyclic state. DASA–PEI characteristic fluorescence, however, is not influenced by these manipulations. On the other hand, the fluorescence of DASA–PEI could be affected by adding Cu^2+^ or Fe^3+^ to an aqueous solution of DASA–PEI. The fluorescence emission of DASA–PEI at 475 nm significantly decreased in response to the metal ion complexation and the detection limits for Cu^2+^ and Fe^3+^ were found to be as low as 1.3 nM and 10.1 nM, respectively. Cu^2+^ sensing was further demonstrated via the “Turn-off” sensing with a pyridine moiety containing DASA [62]. MAFC was hereby reacted with N-((pyridine-n-yl)methyl)ethanamine in THF, resulting in the precipitation of a colorful DASA, which is further designated as PMEA–DASA. Cu^2+^ ions as well as different volatile primary amines were detectable via the loss of coloration due to linear to cyclic isomerization after the addition to the PMEA–DASA molecule. The LOD of Cu^2+^ sensing was demonstrated to be 3.1 µM. Experiments were performed with paper strips coated with PMEA-DASA and revealed that isomerization is reversible as soon as PMEA–DASA is no longer in contact with volatile amines. Further, it was noted that the isomerization process is dependent on the position of the nitrogen atom in the pyridine moiety. The proposed system can be used as a sensitive Cu^2+^ detection agent. These studies have shown that in addition to amine sensing, MAFC can also find potential applications in the development of other sensing devices. The following Table 1 serves as a summarizing overview of all sensors those were reported in this review.

## 5. Outlook

In this review, we have shown that activated furans, especially MAFC, find their prominent position for sensing applications. Hereby, MAFC as a precursor for DASAs gains more and more importance in the field of optical sensing due to its rapid reaction kinetics, easy synthesis and operation, solubility in benign solvents, and high molar absorption coefficient. Even though few publications demonstrated the use of MAFC sensor systems for the detection of metal ions or thermal impact, the focus in this field of research obviously lies in amine sensing. Amines are the most highly consumed industrial material and find applications in various areas from medicine to protein quantification, electrical industry, environmental protection, and food industry. However, the release of volatile or biogenic amines can have adverse health effects when consumed by humans and can further harm the environment. Therefore, the identification of amines in samples is highly relevant. This becomes obvious when looking at the numerous publications and reviews in this field. MAFC as a newly emerging detection agent for amines was so far mostly explored under laboratory conditions. However, some demonstrations of successful real-world sample analyses have been published and we except much more to come. Especially, coated on chips or bound on solid polymer surfaces, MAFC has great potential as a small, cheap, and portable colorimetric sensing system for quick on-site detection of amines. Impressive first results of amine detection with MAFC-coated paper strips in the solution as well as the vapor phase substantiate this claim. Bound in food packaging, MAFC could provide a customer-friendly access to food spoilage status without the need of unpacking a product. In pharmacology, medicine and forensics MAFC is an ideal candidate for the detection of primary amine-containing pharmacological substances, neurotransmitters, or amphetamine-type drugs. Further, we strongly believe that MAFC will play a major role in amino acid and protein quantification in food and biological samples where amines can be detected, e.g., via primary amine residues on amino acid lysine. It was clearly demonstrated that the sensing performance and specificity of MAFC can be influenced via various parameters like solvent polarity, pH, and temperature. Thus, we are expecting much more selective sensing possibilities of MAFC to be revealed in the coming years.

## Figures and Tables

**Figure 1 molecules-28-06627-f001:**
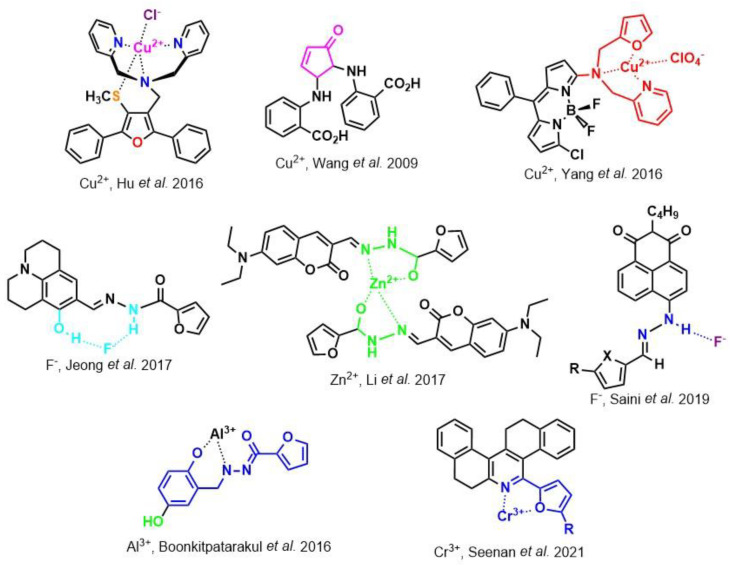
Molecular structure of furan derivative molecular sensors for optical ionic sensing [29,30,31,32,33,34,35,36]. Colored motifs depict the molecular scaffold of the sensor, which is actively participating in the sensing process.

**Figure 4 molecules-28-06627-f004:**
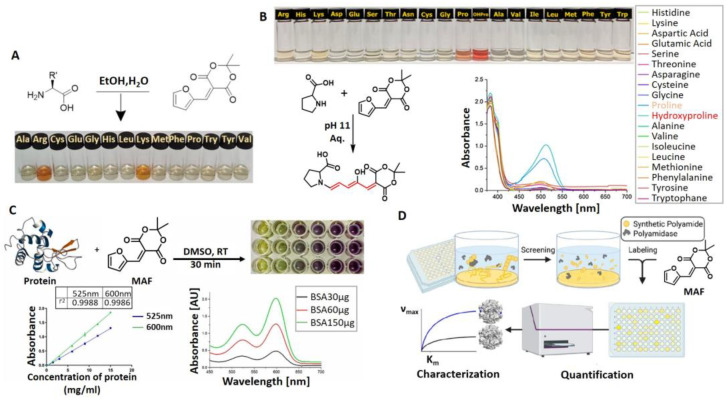
Biologically relevant studies of amino acid (and their structural analogues) and protein sensing with MAFC. (**A**) Visual representation of results from 5 min reaction of MAFC with different amino acids (4 mM). Only lysine and arginine allow DASA formation at physiological pH. Reprinted and adapted from [57]. (**B**) Visual representation of findings of reaction of MAFC with numerous amino acids at pH 11 and corresponding UV-Vis absorption spectra. Distinct color formation is only visible for proline and hydroxyproline. Reprinted and adapted from [58]. (**C**) Overview of findings of protein quantification studies with MAFC as detection agent. Reprinted and adapted with permission from [17]. (**D**) Schematic depiction of high-throughput screening system for optimizations of enzymatic polyamide and polyurethane degradation studies. Reprinted and adapted with permission from [59].

**Table 1 molecules-28-06627-t001:** Overview of all herein reported furan- and activated furan-based sensors.

Activated Furan Species Involved in Sensing	Analyte	Application	LOD	Source
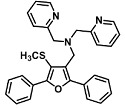	Cu2+	Metal ion chemosensing	0.168 µM	[29]
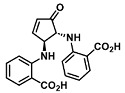	Cu2+	Metal ion chemosensing	15 nM	[30]
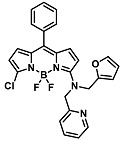	Cu2+	Metal ion chemosensing in living cells	5 µM	[31]
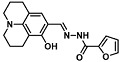	F−	Dental health	10.02 µM	[32]
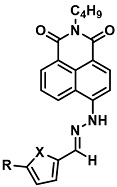	F−, CN-	Sensing of biologically and environmentally pertinent species	<0.3 ppm	[33]
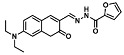	Zn2+	Metal ion chemosensing	0.113 µM	[34]
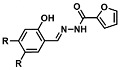	Zn2+, Âl3+	Metal ion chemosensing	3.1 nM	[35]
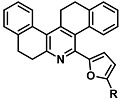	Cr3+	Metal ion chemosensing	142 nM	[36]
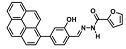	Trinitrotoluene	Identification of explosives	30 µM	[37]
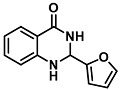	4-Nitrophenol	Environmental protection	16.11 nM	[38]
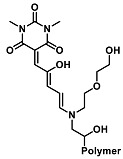	Diethylcyanophosphate	Nerve agent detection	1.0 mM	[46]
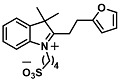	Adrenaline	Neurotransmitter detection	1.37 nM	[47]
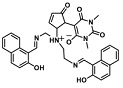	Al3+, Zn2+	Metal ion chemosensing	0.365 µM, 0.1 µM	[18]
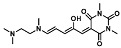	Primary Amines Sc3+, Ti4+, Cr3+, Al3+	Colorimetric sensing of primary amines and high-charged Lewis acids	1.75 µM	[48]
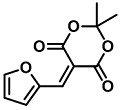	Diethylamine, dimethylamine, piperidine, butylamine, ammonia, cadaverine	Biogenic amine sensing in solution and vapor phase	0.4 ppm, 10 ppm, 10 ppm, 4.4 ppm, 13.2 ppm, 100 ppm	[26]
	2° amines	Amine sensing on polymer surface	na	[52]
	diethylamine, n-butylamine, indoline, p-methoxyaniline	Chemical and thermal amine sensing in aqueous solution	20 ppm, 10 ppm, 20 ppm, 100 ppm	[53]
	Mesalazine	Amine sensing in pharmaceutical products	0.04 µg/mL	[55]
	Methamphetamine, 3,4-methylenedioxymethamphetamine	Sensing of amphetamine-type stimulants	0.36 µg/mL, 0.57 µg/mL	[56]
	lysine	Sensing of amino acids	100 µM	[57]
	Proline, 4-hydroxyproline	Sensing of amino acids	11 µM, 6 µM	[58]
	Bovine Serum Albumin	Quantification of proteins	125 µg/mL	[17]
	Polyamide and Polyurethane degradation products: ’6-aminohexanoic acid, hexamethylenediamine, 2,4-toluenediamine, 2,6-toluenediamine, 4,4′-methylenedianiline	Process control	49 µM, 23.8 µM, 0.6 µM, 10.6 µM, 5.9 µM	[59]
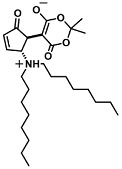		Temperature mapping		[60]
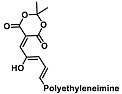	Cu2+, Fe3+	Metal ion chemosensing	1.3 nM, 10.1 nM	[61]
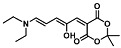	Cu2+	Metal ion chemosensing	100 µM	[49]
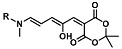	Cu2+	Metal ion chemosensing	3.1 µM	[62]
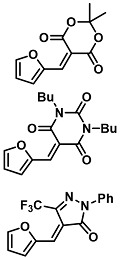	Diethylamine	Volatile amine sensing with electrospun meshes	<1 ppm, 1–10 ppm, 0.1 ppm	[54]

## Data Availability

Not applicable.

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
