# Peer review of "Meldrum’s Acid Furfural Conjugate MAFC: A New Entry as Chromogenic Sensor for Specific Amine Identification"

_molecules, 2023, doi:10.3390/molecules28186627_

Round 1

Reviewer 1 Report

In this manuscript, the author reviewed the “Meldrum’s acid activated furan conjugate MAFCas chromogenic sensor for specific amine identification”.  The manuscript can be publishable in molecules after the following points are fully revised.

1) In Figure 1. "et al." should be in "italic" form, missing hydrogen atoms in ref 25 and ref 26 structures. I also read these original papers.

2) In Figure 4B and 4C, the text is too small to understand. Please make it easier to read.

Author Response

1.) Thank you very much for your encouraging comments and excuse us for overlooking the mistakes of missing hydrogen in Figure 1. The figure is corrected according to the suggestion and et al. has been italicized.

2.) Text and the structure are reformed in Figure 4 for better visualisation. Therefore, in figure caption we have changed our statement from “Reprinted with permission from” to "reprinted and adapted with permission from [ref]”.  

Reviewer 2 Report

This review described Meldrum’s acid activated furan conjugate as an effective probes for the sensing various amines. I suppose this paper might be helpful for many researchers who deals with bioanalytical applications. I recommend its publication in this journal. However, some correction is necessary before being published in Molecules.

1. In the introduction: A significant part of this section (see references 21-29, etc.) is devoted to the detection of various metal cations that is not directly related to the claimed title of the article. On the other hand, the authors practically do not discuss such a hot topic for amine detection as a method based on "turn-on" fluorescent sensors such as those based on small molecules, metal–organic frameworks (MOFs), etc. It is strongly recommended to add some references to this topical analytical approach: https://doi.org/10.1039/D2QM01065H; https://doi.org/10.1016/j.dyepig.2020.108344; https://doi.org/10.1021/acs.inorgchem.7b00787; etc.

2. The authors should make a summarized table that would contain data on the sensors described in the review, the analytes they detect, and the corresponding detection limits.

3. Some figures are too small to see them. Improve the dimensions of each image for a better see.

Minor editing of English language required.

Author Response

1a) In general we agree with the opinion of respected reviewer about the concern of discussing the metal ion detection by furan based sensors in a review where biogenic amine detection is in focus. However, we deliberately, brought this topic in the discussion, because in our opinion, chemists who wish to proceed on the development of activated furan sensors should be made aware about the potential side interactions of furan with other analytes like metal ions. Therefore, we request to keep the discussion in current form.

1b) We thank respected reviewer to highlight the significance of “turn on” amine sensors. We have included suggested reference and some additional literature in the introduction section. In addition, for boarding the scope of amine detection information in our review, we have cited very recent comprehensive reviews as references.

p.2: “Especially for the development of device-based amine sensors an enhancement of fluorescence intensity, so called “turn-on” is desirable. Kumpf et al. has demonstrated the use of extended distyrylbenzenes as stripbased “turn on” assay [19]. Mani et al. developed zinc based coordination polymer for selective “turn on” detection of aliphatic amines [20]. Synthetically simple di-catechol (dicat) have shown to reach the detection limit down to sub ppm level [21]. Similarly, 1,4-diazine-based dyes have shown selective chemosensing ability towards aliphatic amines [22].

2) As suggested, a summary table has been introduced in section 4 of our manuscript.

3) Dimensions of pictures in Figure 3 were adjusted to enhance readability. Text and structure are reformed in Figure 4 for better visualisation. Therefore, in figure caption we have changed our statement from “Reprinted with permission from” to "reprinted and adapted with permission from [ref].